# Mate Perception and Gene Networks Regulating the Early Phase of Sex in *Pseudo-nitzschia multistriata*

Pina Marotta [1,*,†], Camilla Borgonuovo [1,†], Anna Santin [1], Monia Teresa Russo [1], Francesco Manfellotto [1], Marina Montresor [1], Pasquale De Luca [1] and Maria Immacolata Ferrante [1,2,*]

[1] Stazione Zoologica Anton Dohrn, 80121 Napoli, Italy
[2] Oceanography Section, National Institute of Oceanography and Applied Geophysics (OGS), 34151 Trieste, Italy
* Correspondence: pinamarotta82@gmail.com (P.M.); mariella.ferrante@szn.it (M.I.F.)
† These authors contributed equally to this work.

**Abstract:** Diatoms are photosynthetic microorganisms playing a key role in the functioning of aquatic ecosystems; they are at the base of the food web and are the main drivers of biogeochemical processes. These microalgae have a unique diplontic life cycle in which the vegetative phase entails a cell size reduction that would lead to the extinction of the cell population if the size was not restored, usually by sexual reproduction. The switch from asexual to sexual reproduction needs to be finely synchronized and regulated to ensure its success; to this aim, cells evolved complex chemical crosstalk that mediates mating. We focused our attention on the marine diatom *Pseudo-nitzschia multistriata*, investigating the reciprocal perception of the opposite mating type (MT) and the genes and signaling molecules putatively involved in the process. From previously available transcriptomic data, we selected a panel of genes deregulated during the early phase of sexual reproduction, confirming for some of them a role during mate perception and establishing a hierarchy governing their behavior. Moreover, we explored the nature of the molecules controlling sexual reproduction in this species, unveiling that the signaling is mediated by the secretion of protein and non-protein cues from the MT− and MT+, respectively. Our results point to a model in which the two MTs stimulate each other, but the stimulation by the MT− is amplified after MT+ perception in a positive feedback manner.

**Keywords:** sexual reproduction; mate perception; diatom; *Pseudo-nitzschia multistriata*; gene expression

## 1. Introduction

Diatoms, a group of microalgae contributing to ca. 20% of global carbon fixation [1,2], are important contributors to the ocean carbon pump, and form the basis of many aquatic food webs. On the other hand, their excessive proliferation can lead to biofouling or harmful algal bloom events, adversely affecting ecosystems and the economy [3,4]. Moreover, algae are exploited in the biotechnological industry as producers of high-value bioactive compounds [5,6].

Diatoms have a diplontic life cycle, in which a long vegetative phase leads to a cell size reduction until reaching a sexualization size threshold (SST), below which the cells become competent for sexual reproduction, the mechanism by which the original cell size is restored [7,8].

Based on the structure of their cell wall, the frustule, two major groups of species can be recognized: centric and pennate diatoms [9,10]. Centrics have a radial pattern whose outline is often circular, pennates have an elongated shape and are further subdivided into raphid and araphid pennates [10].

The classification in centrics and pennates, generally corresponding to a planktonic and benthic lifestyle (with the notable exceptions of *Pseudo-nitzschia* and *Fragilariopsis*, raphid diatoms with a planktonic lifestyle), also mostly mirrors the split between oogamous and anisogamous sexual reproduction, and homothallic and heterothallic mating systems [11].

One of the main differences between centrics and pennates is that the former engage in meiosis when triggered by environmental cues, while the latter need to encounter a mating partner of the opposite mating type (MT) before starting meiosis (reviewed in Chepurnov et al., 2004 [7]), although external factors (light, temperature, medium composition, and associated bacteria) could have some influence [12,13]. Motility can help in finding the partner: centric diatoms produce motile sperm cells, the presence of pseudopodia has been reported in male gametes of some araphid diatoms, while benthic raphid diatoms can move on the substrate by producing strands of extracellular mucilaginous secretions [14–16]. Within all of these strategies, sex pheromones may assist in finding a mate [7,14,17].

The chemical nature of algal pheromones has been mostly studied in the unicellular green alga *Chlamydomonas reinhardtii* and the multicellular *Volvox carterii* [18–20], as well as in brown algae [15,21–23]; through metabolomic approaches and chromatography assays, highly diverse characteristics and structures ranging from small nonpolar hydrocarbons to high molecular weight polar glycoproteins have emerged [15].

So far, the mate sensing of other diatom species has been investigated, such as in the araphid *Pseudostaurosira trainorii* and in the pennate *Cylindroteca closterium* [17,24], but no compound has been identified yet. The only diatom species in which pheromones have been described is the benthic raphid *Seminavis robusta* [14,16]. Indeed, it has been found that L-proline-derived diketopiperazine (diproline) is released by the mating-type minus (MT−, also conventionally referred to as female strain) after perceiving chemical signals of the opposite MT+ (or male strain). Briefly, when the size of these diatoms is about 50 μm, both MTs constitutively secrete sexual pheromones, SIP+ and SIP-; when MT− perceives its mate through the binding of SIP+, it starts to produce the attraction pheromone diproline; at the same time, MT+ becomes sensitive to diproline and slides toward the diproline source [14,25]. Moreover, by profiling the global gene expression of the cells treated with the pheromone, investigators found that, in addition to the behavioral changes, the mate perception induces remarkable gene expression changes too, that confer to the cells the ability to switch toward sexual reproduction [26].

*Pseudo-nitzschia multistriata* represents one of the experimental models suitable to explore the crosstalk during mate perception and the mechanisms that regulate sexual reproduction. It has a well-described heterothallic mating system [27–29], and numerous resources have been generated in the last few years, including a sequenced genome, transcriptomic data, and genetic manipulation protocols [30–33]. The transcriptional profiling of the two MTs has led to the identification of five MT−related (MR) genes: the MT+ specific genes *MRP1*, *MRP2*, and *MRP3*, and the MT− specific genes *MRM1* and *MRM2*; notably, it has been demonstrated that *MRP3* is the sex-determining gene [33]. Furthermore, the molecular mechanisms regulating the transition from vegetative growth to the onset of sex have been deeply investigated: two opposite MTs were co-cultured for 1 h, 24 h, and 5 days, and the gene expression profile of these phases was characterized through transcriptomics [32]. However, the mate-finding mechanism is still unknown. Experimental data suggest that in *P. multistriata* the switch from the vegetative to the sexual phase is mediated by chemical cues: Basu et al. (2017) [31] cultured MT+ and MT− strains in two bottles separated by a filter that allowed chemical communication but not the contact between opposite MTs. This resulted in the growth arrest of the whole population in the G1 phase, and an important physiological cellular rewiring, as testified by the transcriptomic datasets [31]. However, due to the limitation of the experimental system employed, those experiments provide no clues as to whether a multistage molecular cascade of cause-and-effect interrelation, like that observed in *S. mt* [14,25], is present, and how this is reflected at the molecular level.

Here, exploiting transcriptomic datasets of samples collected during the early phases of sexual reproduction [31,32], we selected a group of genes differentially modulated in the two MTs and, through conditioning experiments in which the cells were placed in the growth medium of the opposite MT, identified those responding directly to primary cues and probably involved in mate perception. These genes could be used as molecular markers to test fractions in a bioassay-guided identification of pheromones involved in the first part

of the mate recognition system. We report experimental evidence for the presence of sex pheromones that reciprocally stimulate the sexualization of compatible strains. The results of our experiments led us to the hypothesis that the two MTs stimulate each other, but the stimulation by the MT− is amplified following exposure to the MT+, in a positive feedback manner. Moreover, we were able to determine that the MT+ putative pheromone is not a protein and has a molecular size below 10 kDa, while the MT− cue could be a protein.

Knowledge of intraspecific signaling of these microalgae may provide insight into the regulation of their life cycle, essential to understanding algal blooms, and offer a tool for manipulating their growth.

## 2. Materials and Methods

### 2.1. Strains Collection and Culture Condition

*Pseudo-nitzschia multistriata* cells were grown in F/2 medium [34] at 18 °C, under white fluorescent light at an irradiance of 60 µmol photons m$^{-2}$ s$^{-1}$ with a 12 h light/12 h dark photoperiod. For the experiments in this study, we used multiple strains, isolated from the LTER station MareChiara in the Gulf of Naples or obtained by crosses carried out in the laboratory (Table S1), to minimize strain-specific effects.

### 2.2. Genomic DNA and RNA Extraction

Genomic DNA was extracted using a modified CTAB extraction procedure. In detail, 25 mL of cells growing in mid-exponential phase were centrifuged for 20 min at 4000× *g*, 4 °C, resuspended in 2 mL F/2 medium, and centrifuged again for 15 min at 20,000× *g*, 4 °C. The resulting pellet was homogenized in 500 µL CTAB extraction buffer (Delchimica Scientific Glassware), then 12 µL of 2-mercaptoethanol (Sigma-Aldrich) and 80 µg/mL of RNAse A (Roche) were added. After 45 min incubation at 65 °C, vortexing every 15 min, and ice shock, 500 µL of chloroform-isoamyl alcohol (24:1 *v/v*) was added. The upper phase was recovered after centrifugation for 10 min at 20,000× *g*, 4 °C, mixed with 1 volume of cold isopropanol (Mallinckrodt Baker), and stored at −20 °C for 1 h. DNA pellet obtained after centrifugation for 30 min at 20,200× *g*, 4 °C, was washed with cold 75% ethanol, through centrifugation for 15 min at 20,000× *g*, 4 °C. The dried pellet was dissolved in 25 µL double-distilled H$_2$O.

RNA was extracted following the published protocol "*RNA extraction from diatom P. multistriata*" (https://www.protocols.io/view/rna-extraction-from-diatom-p-multistriata-261gen627g47/v1 (accessed on 29 November 2022)). Then, RNA concentration was determined using Qubit™ 2.0 Fluorometer (Life Technologies) and RNA quality was checked by gel electrophoresis (1% agarose *w/v*).

### 2.3. Gene Expression Analysis

An aliquot of 0.5 µg of total RNA was reverse-transcribed using the QuantiTect® Reverse Transcription Kit (Qiagen) and used as a template to amplify target genes, using 0.4 µM final concentration of the primers reported in Table S2. The qPCR experiments were performed in technical triplicate through a ViiA 7 Real-Time PCR System (Applied Biosystems) using Fast SYBR Green Master Mix (Applied Biosystems), following manufacturer instructions. The reference genes used in the qPCR were *TUB A* and *CDK-A* [35]. Genes fold expression levels were obtained with the Relative Expression Software Tool-Multiple Condition Solver (REST-MCS) [36].

### 2.4. Capillary Assay

For the investigation of cell attraction, the procedure described in Klapper et al. 2011 [24] was followed. Briefly, two hundred milliliters of the MT− 1334-1 and MT+ 1364-6 (Table S1) exponentially growing cultures were filtered on 0.22 µm Millipore membrane filters (Millipore Corporation—Burlington, MA, USA) and solid phase extracted using Chromabond HLB cartridges (Macherey-Nagel—Darmstadt, Germany) following the instruction manual. MeOH extracts were dried using a rotary evaporator and dissolved in

200 μL $H_2O$; an aliquot of 25 μL of the medium extract was added (1:1 *v/v*) to hot agar (2% Agar–Agar), quickly mixed, and absorbed into the capillaries (l = 3 cm, V = 15 μL, microcaps, Drummond Scientific Company—Broomall, PA, USA) just before use. The air-dried capillaries were then carefully placed vertically in the wells of a 6-well in which cells of the opposite MT were placed. Microscopy observations of the capillary openings were taken at t = 0 min and t = 60 min.

### 2.5. Experiments with Conditioned Culture Media

The conditioning experiments were conducted by incubating MT+ and MT− strains with the filtered growth medium of the opposite MT. Each experiment has been performed in triplicate, on different cultures of the same MT− and MT+ strains (Table S1); all of them were in the exponential phase of their growth, with densities ranging between $2 \times 10^5$ and $3 \times 10^5$ cells mL$^{-1}$. The MT− cells were incubated for 30 min, 1 h, and 2 h with the filtered culture medium of an MT+ strain, while the MT+ cells were incubated for 1 h, 3 h, and 6 h with the filtered culture medium of an MT− strain. These incubation times were found to be the most informative after a series of pilot experiments in which a wider range of timings was tested (data not shown). For each experiment, a co-culture of the same MT pairs investigated was collected at the same time points and used as a control.

Briefly, the conditioned media used for the experiments were obtained by centrifugating culture cells in exponential phase for 15 min at $2000 \times$ *g*, 18 °C, to separate cells from the growth medium; then, the media were further filtered on 0.22 μm Millipore membrane filters (Millipore Corporation—Burlington, MA, USA). Cells to be incubated with the conditioned medium were collected by centrifugation for 15 min at $2000 \times$ *g*, 18 °C, and subsequently resuspended in the filtered conditioned medium (Figure 1a). To assess the strain ability to perform successful crosses, each time a co-culture of the two employed strains was left for an inspection under the microscope after 24 h. The conditioned samples were processed only in case of the presence of sexual stages (pairs, gametes) in this control cross [29].

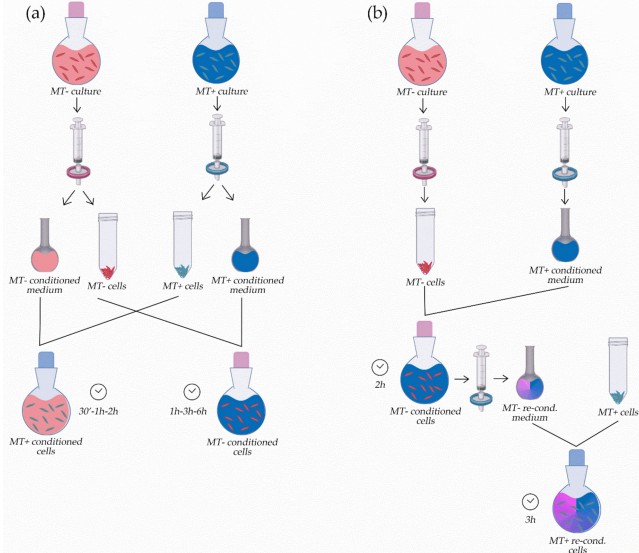

**Figure 1.** Diagram of the conditioning experiments. (**a**) Exponentially growing cultures are used to obtain the conditioned growth medium from the cells. The cell pellet of one mating type (MT) is incubated with the conditioned medium of the opposite one. The conditioned cells are then collected at different time points. (**b**) Diagram of the double conditioning experiment. In the first step MT− cells are incubated with the medium of an MT+ strain for 2 h; next, the same medium used to incubate MT− cells is filtered and used to incubate MT+ cells for a further 3 h.

Double conditioning experiments were performed using the same principle of the protocol described above. Briefly, MT− cells were first incubated for 2 h with the culture

medium of an MT+ strain; then, the incubation medium was collected and filtered again and used to incubate MT+ cells for another 3 h (Figure 1b).

Control cross tests were also set up in parallel in this case.

For heat inactivation, the filtered media were heated for 20 min at 90 °C in a water bath.

### 2.6. Plasmid Construction

To prepare the PmH4p*MRP1*-YFPAt overexpressing vector (vector map: https://zenodo.org/record/3925481#.Y5BnAXbMJPZ (accessed on 29 November 2022)), the primers 13283EcoF and 13283EcoR (Table S2), containing the *EcoRI* restriction site, were used to amplify the *MRP1* coding sequence from the cDNA of the strain Va2CA+, excluding the stop codon, and using Q5® High-Fidelity DNA Polymerase (New England Biolabs) according to manufacturer's instructions. Then, the plasmid PmH4pGTubC-YFP was digested with *EcoRI* at 37 °C in the *EcoRI* buffer (New England Biolabs). The digested plasmid was dephosphorylated using Calf Intestinal (CIP) Alkaline Phosphatase (New England Biolabs). The backbone and the insert were purified with QIAquick™ PCR Purification Kit (Qiagen—Hilden, Germany) and then ligated using the Quick Ligation™ Kit (Qiagen—Hilden, Germany). The vector insert was sequence-verified with Sanger sequencing.

### 2.7. Biolistic Transformation of P. multistriata

*P. multistriata* strains MF1 and MF9 (Table S1) were transformed by microparticle bombardment using the Biolistic PDS-1000/HE Particle Delivery System (Bio-Rad), through the biolistic method [30] recently re-adapted [37] (https://www.protocols.io/view/biolistic-transformation-of-pseudo-nitzschia-multi-bp2l619qzvqe/v1 (accessed on 29 November 2022)). Briefly, gold nanoparticles (0.6 μm diameter, Bio-Rad) were coated with 3 μg of the previously described PmH4p*MRP1*-YFPAt overexpressing vector and 3 μg of the PmH4BleAt plasmid [30,38], using the CaCl$_2$-spermidine method according to manufacturer's protocol, and were shot into $5 \times 10^6$ cells recently plated and dried on 0.4% agarose F/2 medium. After four hours from the shot, cells were gently detached from the solid medium and transferred in a total volume of 200 mL of F/2 medium without selection. After 24 h, 1 μg/mL zeocin was added. Resistant transgenic cells appeared after two-three weeks.

### 2.8. PCR Analysis on Transformed Cells Overexpressing PmMRP1

A first PCR was performed on genomic DNA extracted from transformants and wild-type cells, pelleted from selective and control liquid cultures. Cells were resuspended in 20 μL of a lysis buffer consisting of 1% (*v/v*) Triton X, 20 mM Tris-HCl (pH 8), and 2 mM EDTA (pH 8) in Milli-Q water, vortexed, put on ice for 15 min and then at 85 °C for 10 min, as described in Falciatore et al. (1999) [38]. The diluted solution was then used for PCR screening.

To analyze *PmMRP1-YFP* cassette integration in genomic DNA, primers MRP1fw1 and GFP_down were used (Table S2). Amplification fragments were analyzed on an agarose gel.

## 3. Results

### 3.1. P. multistriata Molecular Response during Mate Perception

In the water column, the ability of a diatom to encounter a mating partner may rely also on the secretion of attractive molecules from one or both the MTs. To assess if the putative pheromones secreted by *P. multistriata* have chemoattractant characteristics, we performed a capillary assay [24], testing the ability of exudates of both MTs to attract the reciprocal partner. To this aim, solid phase extracts, prepared from the growth medium of both MTs, were mixed with agar and used to fill capillaries that were dipped in a well containing cells of the opposite MT; the molecules with attractive capacity were supposed to attract the cells around them. The results of the assay indicated that none of the MTs of this species secretes attracting molecules, capable of guiding the cells to the opposite partner (data not shown).

As an alternative strategy towards the identification of pheromones, we attempted to identify the genes whose induction can be used to track sexualization events. In fact, we hy-

pothesized that when two compatible partners perceive each other, a pheromone-mediated communication is established that triggers the sexualization phase, characterized by the early induction of genes regulating the transition from the vegetative stage to meiosis. The results of the transcriptomic experiments illustrated by Basu et al., 2017 [31] and Annunziata et al., 2022 [32] provided a snapshot of the genes regulated during the early phase of sexual reproduction, among which it is possible to find candidate genes directly induced by the chemical cues of the opposite MT. Therefore, we exploited these findings by selecting the most differentially expressed genes in both datasets in a concordant manner (Table 1). Interestingly, among the most upregulated genes in MT− we found the endopeptidase Cathepsin D, the MT− related *MRM2* [33], and the meiotic gene *Rad51A1* [39]; moreover, we found the uncharacterized gene 7488, whose in silico analysis revealed the presence of a signal peptide in the primary sequence (Figure S1). On the other hand, the MT+ related *MRP1* [33] is the most overexpressed gene in MT+ during the sexualization. Hence, we tested if these genes participate in the opposite MT's molecular response when exposed to the putative chemical signals constitutively secreted in the medium by the opposite MT. To this aim, we set up a "conditioning system" that mimics the sexualization of cells. In this setup, the growth medium of the cells of a given MT is completely replaced with the growth medium of the opposite MT (Figure 1a). After incubation, the cells are collected and analyzed through qPCR (Figure 2).

**Table 1.** List of the most differentially expressed genes selected to study mate perception in *Pseudonitzschia multistriata*. For each of them the corresponding level of induction in sexualized strains at the time points 2 h and 6 h, and in co-cultures at the time point of 1 h, 24 h, and 5 days (d), are indicated [31,32].

| Gene Name | Gene ID | Log2 Fold Change (Basu et al., 2017 [31]) | | | | Log2 Fold Change (Annunziata et al., 2022 [32]) | | | | | |
|---|---|---|---|---|---|---|---|---|---|---|---|
| | | MT+ | | MT− | | Cross vs. MT+ | | | Cross vs. MT− | | |
| | | 2 h | 6 h | 2 h | 6 h | 1 h | 24 h | 5 d | 1 h | 24 h | 5 d |
| Cathepsin D | PSNMU-V1.4_AUG-EV-PASAV3_0103000.1 | −0.4 | 0.5 | 5.3 | 7.0 | 9.2 | 8.5 | 8.2 | 5.1 | 4.4 | 4.1 |
| MRM2 | PSNMU-V1.4_AUG-EV-PASAV3_0006960.1 | 0.5 | 0.1 | 1.8 | 2.5 | 9.1 | 8.8 | 8.0 | 1.2 | 1.0 | 0.0 |
| 7488 | PSNMU-V1.4_AUG-EV-PASAV3_0020420.1 | 0.4 | −0.2 | 5.7 | 6.8 | 8.0 | 7.0 | 6.6 | 4.8 | 3.8 | 3.4 |
| Rad51A1 | PSNMU-V1.4_AUG-EV-PASAV3_0056780.1 | 0.8 | 5.8 | 1.8 | 4.8 | 6.1 | 7.8 | 5.2 | 6.2 | 8.0 | 5.3 |
| MRP1 | PSNMU-V1.4_AUG-EV-PASAV3_0024820.1 | −0.1 | 3.7 | −0.1 | −0.9 | 2.5 | 6.0 | 4.4 | 3.7 | 7.4 | 5.7 |

-10      0      10 The color pattern indicates the Log2 Fold Change, ranging from −10 (red) to +10 (green).

Analyzed with respect to the untreated MT− cultures, the expression of Cathepsin D, *MRM2*, 7488, and *Rad51A1* was induced following exposure to the culture medium of the opposite MT alone (Figure 2a–d). Moreover, their upregulation was comparable to and had the same trend as that obtained during a co-culture setup (Figure 2a–d), with 7488 and Cathepsin D among the most upregulated genes (Figure 2a,d). The specular experiment performed on MT+ strains also resulted in the induction of the *MRP1* gene with a peak of expression after 3 h of incubation (Figure 2e).

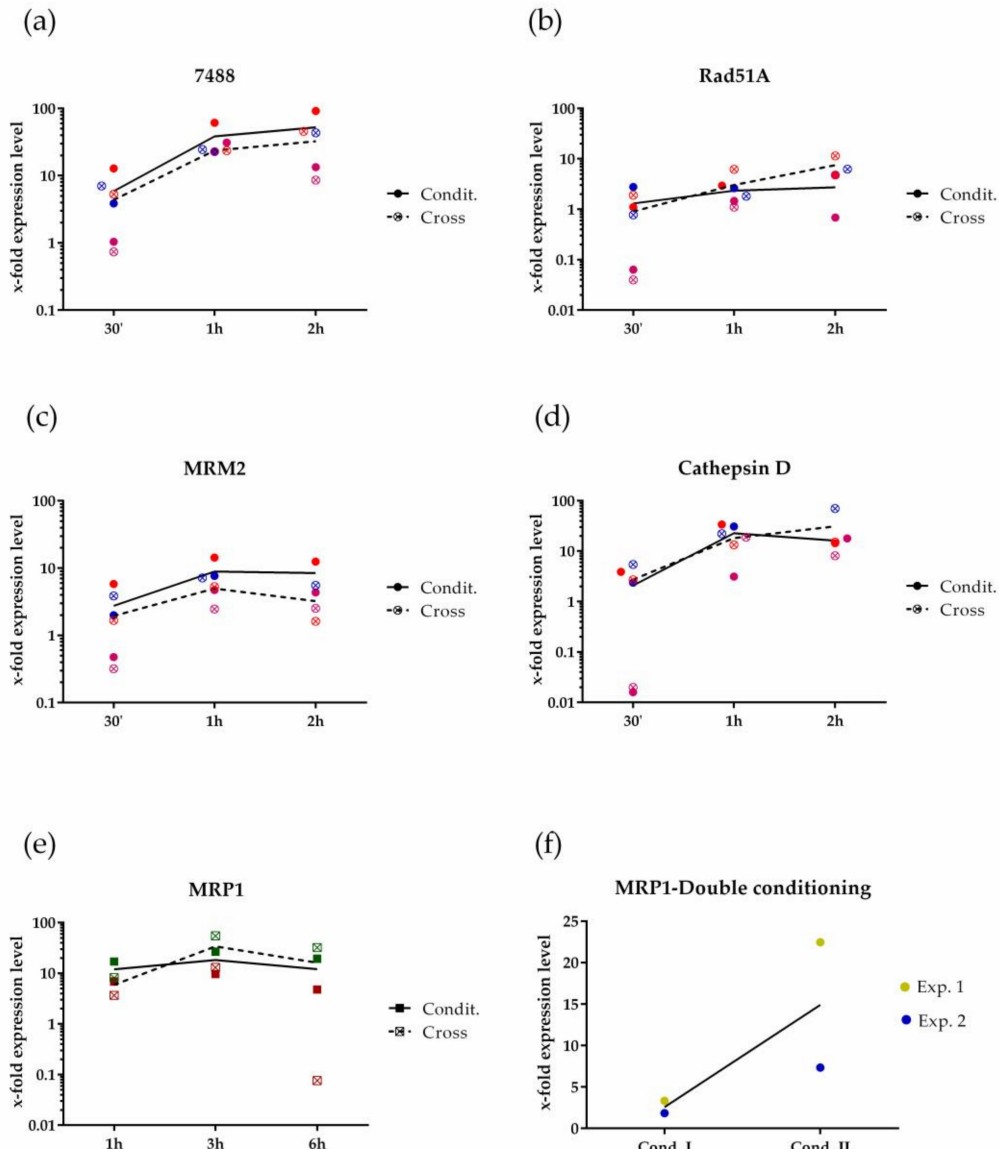

**Figure 2.** Gene expression analysis in experiments with medium conditioned by the opposite MT. Time course of the expression levels of sexualization markers in *P. multistriata* cells incubated with the filtered medium of the opposite MT culture and comparison with the expression levels in the corresponding co-cultures where sexual reproduction occurs. (**a–d**) 7488, *Rad51A*, *MRM2*, and Cathepsin D qPCRs performed on different MT− strains "conditioned" (closed circle) or co-cultured (crossed circle) with different MT+ strains; (**e**) *MRP1* qPCR performed on different strains of MT+ "conditioned" (closed square) or co-cultured (crossed square) with different MT− strains. Circles and squares represent the fold changes obtained in distinct experiments with different couples of opposite MTs (the same color corresponds to the same experiment); lines represent the median of the different experiments (continuous for the conditioning experiments, dashed for the co-cultures). (**f**) *MRP1* qPCR performed on MT+ "single-conditioned" (Cond. I) or "double-conditioned" (Cond. II). Dots represent the expression levels obtained in distinct experiments with different couples of opposite MTs (the same color corresponds to the same experiment); lines represent the median of the different experiments. In each experiment, the expression levels are expressed as x-fold with respect to the corresponding untreated strain, in a logarithmic scale from (**a**) to (**e**), in a linear scale in (**f**).

On the other hand, considering the RNA-seq datasets [31,32], the induction of *MRP1* in the MT+ cells was temporally delayed compared to the two most-induced genes in MT− cells, 7488 and Cathepsin D. Hence, we tested the hypothesis that the female cells increased

the secretion of their sexual cues (or produced new ones) when they perceived a male, with the effect of boosting the sexual response of the male. To this aim, we performed a "double conditioning" experiment: MT− cells were incubated with the culture medium conditioned by an MT+ strain for 2 h, and then this medium was used to incubate MT+ cells for 3 h (Figure 1b). In doing so we expected to enrich the conditioned medium with the boosting female factor/s secreted by MT− once they sensed a mate. As illustrated in Figure 2f, this experiment was performed twice, using different *P. multistriata* strains (Table S1). MT+ cells were treated with both a normal MT− culture medium (single-conditioning) and the "boosted" one (double-conditioning), and after each incubation, cells were collected, and the RNA extracted. Compared to the untreated cells, the *MRP1* expression levels of the samples incubated with the "boosted" medium were higher than the corresponding single-conditioned ones.

To expand the list of sexualization markers in Table 1, we tested other differentially expressed genes (Table S3). By qPCRs carried out on monocultures of MT− and MT+ strains, we first tested if these genes were more expressed in one of the two MTs, suggesting a potential role in the process of sex specification. We found that 77,510, Calmodulin, 29,310, and 67,000 displayed a variable marked overexpression in control MT+ vs. control MT−, while only *Fas1* seemed to be more expressed in the control MT− with respect to the control MT+ (Figure S2a). Then, to test if these genes are induced during sexual reproduction, their expression was measured in strains co-cultured for 2 h with the opposite MT, but none of them were upregulated (Figure S2b,c). Therefore, we decided to continue with *MRP1* and the most upregulated genes 7488 and Cathepsin D in the next experiments.

### 3.2. Profiling of P. multistriata Putative Sex Pheromones Properties

To identify the molecules responsible for mate perception and for triggering the downstream signaling cascade leading to meiosis, first, we focused on *MRP1*, identified by Russo et al., 2018 [33] as MT+ specific. This gene is strongly upregulated during sex, but its function is unknown. Its primary sequence contains a putative signal peptide that suggests the possibility that this protein of about 19.4 kDa could be secreted and active as a sex pheromone [33]. To test this hypothesis, we produced the cleaved *MRP1* protein in a bacterial heterologous system, to test its effects on *P. multistriata* cell cultures. Three different concentrations (100 ng/mL, 1 μg/mL, and 10 μg/mL) of the heterologous *MRP1* protein were tested on MT− cells to assess its ability to stimulate changes in the expression levels of 7488 and Cathepsin D, but no effects were found (data not shown).

To determine the molecular size of the putative pheromone, the growth medium of an actively growing MT+ strain was fractionated through ultrafiltration columns, to obtain fractions containing molecules with a different range of molecular weights: above 30 kDa, between 30 and 10 kDa, below 10 kDa. These fractions were tested for their ability to induce the MT− sexualization markers, 7488 and Cathepsin D, after MT− cells have been exposed to them for 2 h, as the "conditioning" experiment described in the previous paragraph. A preliminary experiment was carried out to determine if the active fraction was above or below 30 kDa, finding that it was below (Figure 3a). Next, the fractions between 30 kDa and 10 kDa, and below 10 kDa were tested. The four replicates of this experiment, which have been performed on different strains to account for the system's biological variability (Figure S3a), demonstrated that the fraction below 10 kDa was more active. We, therefore, concluded that the size of the MT+ cue is presumably below this molecular weight (Figures 3b and S3a) and excluded the hypothesis that the cleaved form of *MRP1*, of about 19.4 kDa, could act as a pheromone.

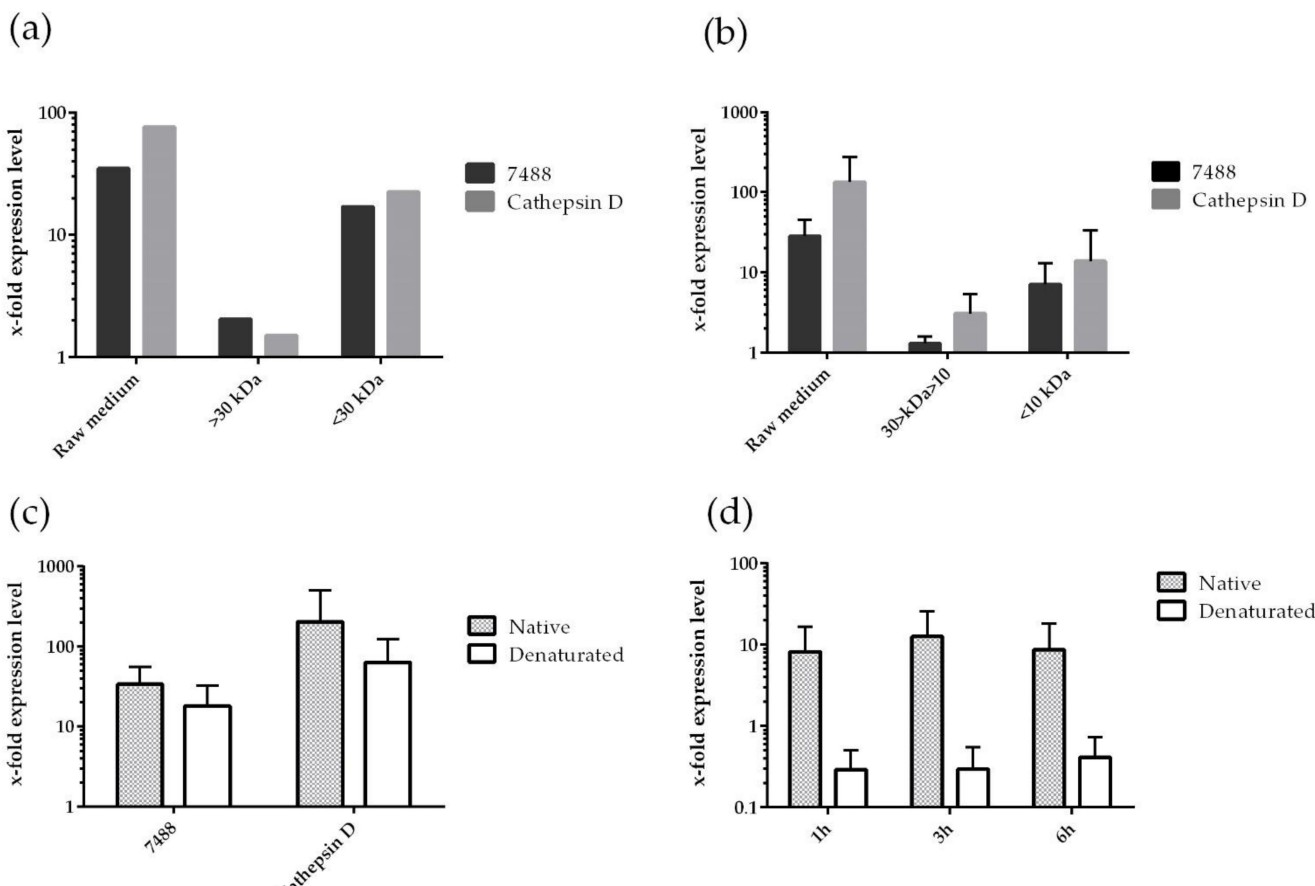

**Figure 3.** Chemical-physical properties of *P. multistriata* putative pheromones. (**a**) 7488 and Cathepsin D qPCR performed on MT− cells conditioned with the raw medium and the fractions above and below 30 kDa of an MT+ culture medium; the bars represent a single experiment. (**b**) 7488 and Cathepsin D qPCR performed on MT− cells conditioned with the raw medium and the fractions between 30 and 10 kDa and below 10 kDa of an MT+ culture medium; means ± SD (*n* = 4, with different strains). (**c**) 7488 and Cathepsin D qPCR performed on MT− cells conditioned with the native and the heat-inactivated raw medium of an MT+ culture medium; means ± SD (*n* = 7, with different strains). (**d**) *MRP1* qPCR performed on MT+ cells conditioned with the native and the heat-inactivated medium of an MT− culture medium; means ± SD (*n* = 2, with different strains). Each expression value is normalized for the correspondent MT− or MT+ untreated cells.

Furthermore, to test if the MT+ and MT− cues were of a protein nature, we heated for 20 min, at 90 °C, the culture medium of each MT to denature any secreted proteins and used it to incubate the opposite MT (Figures 3c,d and S3b,c). Compared to the treatment with the untreated medium, in MT− we did not observe any conspicuous difference in the response of 7488 and Cathepsin D to the heat-inactivated medium (Figure 3c). In the MT+, we used the *MRP1* induction as a read-out of the response of MT+ cells to the MT− cues. We found that there was no *MRP1* overexpression in MT+ cells conditioned with the MT− heated growth medium (Figures 3d and S3c). From these results, we concluded that the MT− cue may be a protein, while the putative MT+ pheromone is not unless its molecular weight is so small to make any heat treatment ineffective.

*3.3. MRP1 Functional Analysis*

To further investigate the *MRP1* function we opted for an ectopic expression of the gene in MT− cells. To this aim, the CDS of *MRP1*, amplified by RT-PCR from an MT+ strain, was cloned downstream of the constitutive *P. multistriata* histone H4 promoter [30], in frame with the yellow fluorescent protein YFP (Figure S4a). Two MT− strains, MF1 and MF9,

were co-transformed via particle gun bombardment with the plasmids MRP1-YFP and pFCPBp-Sh ble [38], which contains the zeocin resistance gene. Integration of the transgene in the resistant strains was verified by PCR on the genomic DNA. Out of 11 resistant transformants, 5 were positive, and 4 of them displayed the expression of the exogenous gene by RT-PCR (overexpressing (oe) clones A1, B1, and 2 from the MF1 strain, clone A20 from the MF9 strain) (Figure S4b).

To quantitatively rate the expression of *MRP1* in the four transformants, we measured the transcription levels of the exogenous gene by qPCR (Figure 4a). This analysis showed a wide range of fold change (FC) with respect to the expression value of the respective parental strain, ranging from 4,4 in the oe-B1 clone to about 180 of oe-A20 (Figure 4a). Despite the high expression level of the exogenous gene, observations by confocal microscopy did not reveal any fluorescent signal, that we expected to observe by virtue of the fluorescent tag; this may be due to incorrect folding effects sometimes found in fusion proteins.

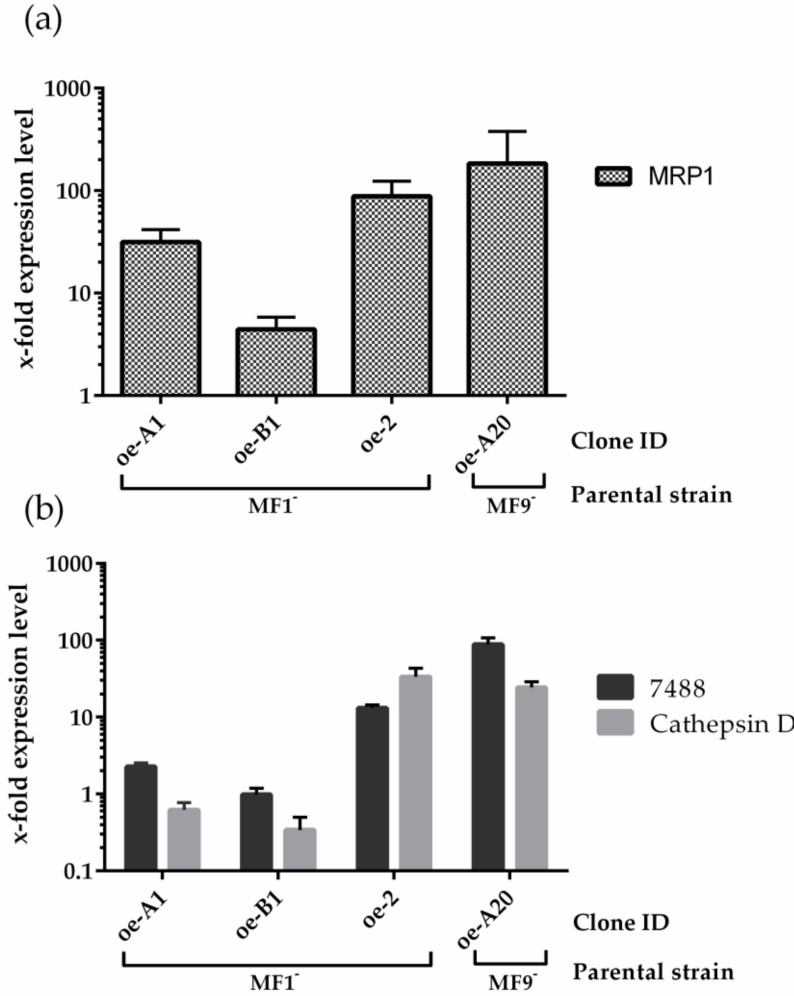

**Figure 4.** Characterization of the MRP1-overexpressing MT− cells. (**a**) MRP1 qPCR performed on the transformed clones. (**b**) 7488 and Cathepsin D qPCR performed on the 4 transformed clones. means ± SD (*n* = 2).

All these transformants, when tested for their ability to mate, produced sexual stages only if crossed with opposite MT+ strains. However, to determine if the ectopic expression of the MT+-specific gene *MRP1* in MT− cells can induce self-sexualization, we checked the expression level of the markers 7488 and Cathepsin D (Figure 4b). The qPCR performed on the four transformants revealed that the induction of these two genes appeared directly correlated to the expression of *MRP1*; in fact, the transformants oe-2 and oe-A20, in which *MRP1* had the highest levels of upregulation, showed the highest 7488 and Cathepsin levels.

Hence, we wondered if these transformants were able to commit sexualization in wt MT− strains. This ability was assessed by performing conditioning experiments in which the growth medium of the oe strains was used for the incubation of female cells for 2, 5, and 24 h, but no effects were observed on the expression of 7488 and Cathepsin D genes (data not shown). These findings reveal a role of *MRP1* in the sexualization process of the MT− cells, by inducing the overexpression of the sexualization markers, but it seems not to be sufficient to stimulate another strain nor to induce gametogenesis.

## 4. Discussion

We investigated the molecular mechanisms involved in the mate sensing process of the planktonic pennate diatom *P. multistriata*, trying to answer the question of how planktonic cells can signal to each other to pair in the water column. In the diatom species investigated so far, sex pheromones trigger several "easy-to-follow" responses that can be helpful to conduct bioassay-guided fractionations, such as induction of gametogenesis and attraction of the opposite MT [17,24,25]; it is not the same as for *P. multistriata*.

When two complementary strains of *P. multistriata* are cultured in an apparatus that impairs the physical contact but allows the exchange of molecules, cells change their transcriptional profiles in response to the perception of the chemical cues deriving from the mating partner [31], putative sex pheromones able to induce mitotic arrest and stimulate the upregulation of meiosis-related genes in the co-cultured strains [31]. On the other hand, morphological evidence for the formation of gametes was never obtained in strains cultivated in such apparatus, demonstrating that physical contact between the two opposite MTs is required for the formation of gametes, unlike the araphid pennates *P. trainorii* and *Tabularia fasciculata* [17,40], in which gametes can differentiate directly in response to the chemical cue of the opposite MT.

We then evaluated the chemoattractant power of the exudates of the two MTs. By capillary assay, we showed that, unlike *S. robusta* and *C. closterium* [24,25], none of the two MTs secrete attracting pheromones; this is in agreement with our previous observations and with the interpretation made in Scalco et al., 2015 [29], according to which, in culture, both MTs actively search for a partner through a series of apparently random movements that bring compatible partners into contact [29].

In the lack of a clear phenotypic assay to test MT perception, we tested if a panel of genes activated during this process may be used as molecular marker for tracking sexualization events. We exploited the RNA-seq experiments performed on *P. multistriata* during the early phase of sexualization and the initial phase of sexual reproduction [31,32], which depict the transcription profile of the cells just before sex takes place. We selected the most upregulated genes in both these datasets (Table 1), appointing 7488, Cathepsin D, *Rad51A1*, and *MRM2* as the molecular proxies of sexualization in MT−, and *MRP1* in MT+.

Thus, we analyzed if and how the expression of the above defined sexualization markers changes in response to the exposure of the opposite MT culture medium alone (Figure 2a–e), demonstrating they are actually induced, comparably to a cross. We can thus conclude that the perception of the chemical cues released by the mating partner is sufficient to trigger the sexualization process in both MTs, even if the response to the constitutively secreted MT− cue was not very strong.

Indeed, activation of 7488 and Cathepsin D in MT− occurs earlier than activation of MRP1 in MT+ [31,32] (Table 1), we thus carried out a "double conditioning" experiment, in which MT− and MT+ cells were sequentially sexualized (Figure 2f). These experiments showed that, when co-cultured, the two MTs start to exchange chemical signals in a sequential manner, where MT− cells, sexualized by the MT+ pheromones, increase the amount of secreted sexual signals to boost the MT+ sexual engagement. This behavior is in agreement with the global gene expression profile observed in the experiment of Basu et al. [31], in which the number of regulated genes is higher in MT− compared to MT+, and with the microscopy observations made in a study by Scalco et al. [29], showing that MT− cells undergo meiosis 30 min earlier than MT+ cells. All these results may be explained

by a signaling cascade relying on multiple pheromones produced by the two MTs, as demonstrated for *S. robusta* [25] and postulated for *P. trainorii* [17] and *C. closterium* [24], probably evolved to ensure the finding of a partner to pair with, so that the energetically expensive sex can start.

The presence of a putative signal peptide in the primary sequence of MRP1, its relatively small size, and the high expression level of this gene in MT+ cells during mate perception, led us to hypothesize that it might be the MT+ pheromone, as in several species of the green algae *Volvox* and *Closterium*, in which peptidic pheromones have been found and characterized (extensively reviewed in Frenkel et al., 2014 [15]). However, the recombinant cleaved form of the MRP1 protein did not induce sexualization of MT− cells (data not shown), and the size fractionation experiments performed on the culture medium of MT+ strains demonstrate that the putative sex pheromone is below 10 kDa, ruling out MRP1 (19.4 kDa) as the putative candidate (Figure 3a,b). Moreover, following the conditioning of MT− cells using the MT+ heat-denatured growth medium, the induction levels of 7488 and Cathepsin D were comparable to those of the cells incubated with the untreated medium; thus, we conclude that that the MT+ cue is not a protein (Figure 3c).

Nonetheless, *MRP1* still remains a good candidate as a key regulator of the MT− sexualization mechanism; in fact, we demonstrated that the ectopic expression of this gene in MT− cells induces, almost proportionally, the overexpression of the MT− sexualization markers, 7488 and Cathepsin D (Figure 4b).

All the findings described in this work give useful information that allows to advance in the understanding of the molecular mechanisms that regulate sexual reproduction in *P. multistriata*. The set of genes induced in this species during the early phase of sexual reproduction was successfully used to track the sexualization events during "conditioning" experiments, that helped to partially characterize the pheromones properties in this species. The MT+ cue has a size below 10 kDa; it might be a metabolite produced by a secondary pathway, or it might be a very small peptidic molecule since its action is not affected by heat treatments. On the contrary, the MT− cue could be a protein because the heat affects its ability to induce *MRP1* in MT+ cells (Figure 3d); moreover, its sexualization power increases (or is supported by newly available molecules) when the MT− cells perceive their partners (Figure 2f).

*MRP1* was already identified as MT+ related gene [33] and, here, we placed it upstream of the pathway leading to the MT− sexualization, as demonstrated by the increased expression of 7488 and Cathepsin D in the transformant clones (Figure 4b). 7488 is an unknown protein whose signal peptide could hint at an extracellular localization (Figure S1); on the other hand, Cathepsin D is a peptidase that in *Saccharomyces cerevisiae* degrades the pheromone to generate a gradient useful for the detection of the nearest mating partner [41]. However, the knowledge we have so far only sketch the rather intricate mechanism underlying the reciprocal recognition of two mating partners and the successive sexual commitment in *P. multistriata*. Nevertheless, the data produced in this paper will facilitate the work needed to identify the chemical nature of *P. multistriata* pheromones. The marker genes identified, and the experimental setup here presented, now enable bioassay-guided fractionation experiments in *P. multistriata*. Moreover, exometabolomic and exoproteomic experiments could be designed to profile the secreted molecules exclusive of each MT. Proteomic studies specifically could help to test the hypothesis that the MT− pheromone is a protein. Finally, a detailed characterization of the sex related genes used in this study, by gain and loss of function strategies, through the generation of overexpressing or knock-out strains, will help to reconstruct the gene networks and the molecular cascade involved in the response to sexual cues. Altogether, these studies will shed light on how microorganisms can communicate in the sea, as well as expanding current knowledge of chemical diversity of these microorganisms.

**Supplementary Materials:** The following supporting information can be downloaded at: https://www.mdpi.com/article/10.3390/jmse10121941/s1, Figure S1: Outputs of prediction software for transit peptides for the 7488 protein; Figure S2: Identification of MT−specific sexualization markers; Figure S3: Chemical–physical characterization of P. multistriata pheromones; Figure S4: Scheme of the MRP1 construct and screening of the resistant clones; Table S1: List of the strains used; Table S2: List of the primers used; Table S3: Complete table of all the target genes.

**Author Contributions:** Conceptualization, P.M., C.B., M.T.R., M.M. and M.I.F.; methodology, P.M., C.B, A.S., M.T.R., P.D.L. and M.I.F.; validation, P.M. and F.M.; formal analysis, P.M.; resources, P.D.L. and M.I.F.; writing—original draft preparation, P.M.; writing—review and editing, P.M., A.S., M.T.R., F.M., M.M. and M.I.F.; supervision, M.I.F.; funding acquisition, P.D.L. and M.I.F.; investigation, P.M., C.B., A.S. and M.T.R. All authors have read and agreed to the published version of the manuscript.

**Funding:** This research was funded by the Gordon and Betty Moore Marine Microbial Initiative, grant number 7978. P.M. was supported by a research fellowship funded by the grant PON PRIMA—Rafforzamento Del Capitale Umano, n. CIR01_00029. C.B. and A.S. were supported by a PhD fellowship funded by the Open University—SZN PhD program.

**Institutional Review Board Statement:** Not applicable.

**Informed Consent Statement:** Not applicable.

**Data Availability Statement:** Not applicable.

**Acknowledgments:** The authors thank Rossella Annunziata for her assistance and suggestions in setting up the experiments, Carmen Minucci and Ferdinando Tramontano for their assistance in cell cultures, and Elvira Mauriello and Raimondo Pannone of the Sequencing and Molecular Analysis Center (SMAC).

**Conflicts of Interest:** The authors declare no conflict of interest.

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
