# Peer review of "Mate Perception and Gene Networks Regulating the Early Phase of Sex in Pseudo-nitzschia multistriata"

_jmse, doi:10.3390/jmse10121941_

Round 1

Reviewer 1 Report

The manuscript is a study on the chemical and molecular evidence of substances involved in the sexual reproduction of the marine diatom Pseudo-nitzschia, a type of diatom well known as a recently toxic diatom. Among the infochemicals, kairomones, and pheromones known as chemical signaling substances between organisms, the study of molecular regulation of substances related to sexual reproduction of microalgae, such as diatom, is particularly interesting. Please add a literature review of similar studies (other biota - especially microalgae). According to the experimental results, further research is needed, but it is judged to be very valuable as basic data for the next study.

Here are some detailed comments.

1. For the reproduction experiments of other researchers, a more detailed explanation of the “experimental hypothesis and process” in the term “one or double conditioning experiment” is needed.

2. Similarly, capillary assays also require detailed explanations with pictures.

3. Further explanation is needed on the effect of the diatom of the experimental organism (because it is a culture strain) on this experiment.

4. Experimental results (number of samples) - The standard deviation of the graph and supplemental data in the text is relatively uniform. SE? SD? Please explain.

5. Please explain and add to the text what the experimental content that the authors want to prove in the next related research.

Reviewer 2 Report

The manuscript by Marotta, P. et al investigate the reciprocal perception of the opposite mating type and gene network regulation in the early phase of sex in a marine diatom P. multistriata. The manuscript sounds interesting and written well; however, a minor revision is required before getting accepted.

The introduction section should be revised with updated references and reduced text as much as possible. The introduction is unnecessarily large but without updated findings.   
